Living on the edge: urban fireflies (Coleoptera, Lampyridae) in Morelia, Michoacán, Mexico

http://orcid.org/0000-0002-6698-2524 Pérez-Hernández Cisteil X. 1 2
Gutiérrez Mancillas Ana María 3
http://orcid.org/0000-0003-3862-1024 del-Val Ek 4
http://orcid.org/0000-0002-2749-8369 Mendoza-Cuenca Luis 2 lfmendoza@umich.mx
1 IUCN SSC Firefly Specialist Group, Gland, Switzerland , Gland , Switzerland
2 Faculty of Biology, Behavioral Ecology Laboratory, Universidad Michoacana de San Nicolás de Hidalgo , Morelia, Michoacán , Mexico
3 Universidad Autónoma de Occidente, Los Mochis, Sinaloa, México , Los Mochis, Sinaloa , Mexico
4 Instituto de Investigaciones en Ecosistemas y Sustentabilidad, Universidad Nacional Autónoma de México , Morelia, Michoacán , Mexico
Silva Daniel
Electronic publication date: 2023 Dec 14
Publication date: 2023
Volume: 11
Electronic Location ID: e16622
Received 2023 Aug 23; Accepted 2023 Nov 16
Copyright: © 2023 Pérez-Hernández et al.
Copyright year: 2023
Copyright holder: Pérez-Hernández et al.
License: This is an open access article distributed under the terms of the Creative Commons Attribution License, which permits unrestricted use, distribution, reproduction and adaptation in any medium and for any purpose provided that it is properly attributed. For attribution, the original author(s), title, publication source (PeerJ) and either DOI or URL of the article must be cited.
License URL: https://creativecommons.org/licenses/by/4.0/

Keywords: Urbanization, Local extinction, Nocturnal insects, Peri-urbanization, Light pollution, Urban green areas, Citizen science, Photinus, Photuris, Conservation areas

Funding: Coordination of Scientific Research, Universidad Michoacana de San Nicolás de Hidalgo (UMSNH) Institute of Science, Technology and Innovation of the Government of Michoacán Program for Scientific Research Projects of Regional Impact PICIR-071 The National Council of Science and Technology (CONAHCyT) funded Cisteil Xinum Pérez-Hernández’s Postdoctoral Project through the program Estancias Posdoctorales por México Universidad Michoacana de San Nicolás de Hidalgo Institute of Science, Technology and Innovation of the Government of Michoacán Program “Comparte tus Ideas” ICTI/D.A./213/2023 Coordination of Scientific Research, Universidad Michoacana de San Nicolás de Hidalgo Entomological sampling for this work was funded by the Coordination of Scientific Research, Universidad Michoacana de San Nicolás de Hidalgo (UMSNH) and the Institute of Science, Technology and Innovation of the Government of Michoacán, through the Program for Scientific Research Projects of Regional Impact (PICIR-071). The National Council of Science and Technology (CONAHCyT) funded Cisteil Xinum Pérez-Hernández’s postdoctoral project, of which this publication is a part, through the program “Estancias Posdoctorales por México”. This work was published using the annual institutional membership of the Universidad Michoacana de San Nicolás de Hidalgo through the joint support of the Institute of Science, Technology and Innovation of the Government of Michoacán, through the Program “Comparte tus Ideas” (ICTI/D.A./213/2023) and the Coordination of Scientific Research, Universidad Michoacana de San Nicolás de Hidalgo. The funders had no role in study design, data collection and analysis, decision to publish, or preparation of the manuscript.

==============================
Fireflies (Coleoptera, Lampyridae) are a globally threatened group of insects due to habitat loss and fragmentation, light pollution, climate change and pesticides. However, against all odds, some firefly populations persist in urbanized environments where all four of these factors are present simultaneously. In this work, we compiled several data sources to document the diversity of fireflies in the urbanized area of Morelia, characterize their current habitats, and determine the main stressors affecting these bioluminescent insects. We found seven genera and 26 species of fireflies (19 nocturnal, seven diurnal) associated with 32 urban, peri-urban and extra-urban areas; at least, 14 are new records for Michoacán, and the list for the state now includes nine genera and 41 species. Five additional sites were documented as extinction sites. We compared the characteristics of these five sites with those of the sites with extant populations. We found that in Morelia, fireflies are mainly associated with areas that have high to moderate proportions of vegetation cover, are near water bodies, have very gentle to moderate slopes, and are exposed to low levels of light pollution. In contrast, the extinction sites showed high proportions of artificial surfaces and high levels of light pollution. Because some fireflies are considered bioindicators of ecosystem integrity as they are associated to specific habitats, are highly diverse and due to their sensitivity to environmental changes, we consider that sites from Morelia’s urban core and extinction sites show the highest levels of environmental degradation, threatening most fireflies and other insects living in the urban core with local extinction. At the same time, our results also suggest that implementing conservation strategies and sustainable planning for the urban development of Morelia in the short term could allow fireflies and other vital elements of the city’s insect communities to persist for future generations. Restoration and conservation of green areas and nighttime environments are essential for biodiversity and human health, especially in intra-urban zones.

Introduction

Urban areas are geographical barriers for biodiversity. The most strongly modified environments (i.e., the intra-urban or urban core areas) are separated from less drastically transformed suburban, rural, agricultural and wildland zones (extra-urban areas) by an ecotone or peri-urban zone that can sometimes act as continuum, blurring the boundary between urban and non-urban (MacGregor-Fors, 2010). Peri-urbanization refers to the increasing phenomenon of accelerated urbanization in this zone at the periphery of the cities that implies land-cover change, increasing building infrastructure and ecological disturbance, which impacts natural ecosystems and biodiversity, and modifies the urban lifestyle of human communities (Viviani, Rocha & Hagen, 2010; Ortiz & Vieyra, 2018; Tzortzakaki et al., 2019).

Some studies have shown that high levels of urbanization change insect community structure and provoke a severe decline in their abundance, species richness and diversity (Merckx & Van Dyck, 2019; Tzortzakaki et al., 2019). Insect communities associated with urban core zones also tend to be more homogeneous and less diverse in their ecological functions, which increases their susceptibility to extinction (Rocha-Ortega & Castaño-Meneses, 2015; Merckx & Van Dyck, 2019). Therefore in peri-urban areas, insect diversity can be higher than within the urban core. This is probably due to the presence of both native and introduced entomofauna and of generalist insects that are able to persist within the more diverse/less modified remnant vegetation patches of peri-urban areas (López-López, 2011; Tzortzakaki et al., 2019).

Recently, it has been stated that firefly populations are declining globally due to factors associated with human activities (Picchi et al., 2013; Lewis et al., 2020; Fallon et al., 2021). Habitat loss, light pollution, and climate change are the main current factors threatening firefly populations, and these factors are also associated with urbanization, industrialization, and agricultural intensification within cities and surrounding areas (De Cock, 2009; Hagen et al., 2015; Fallon et al., 2021; Vaz et al., 2021). Fireflies are usually associated with well-preserved terrestrial ecosystems, from temperate forests to deserts and marshes (Fallon et al., 2021). They occupy different microhabitats during their life cycle; larvae and pupae can be terrestrial, aquatic, semiaquatic, fossorial, arboreal, or scansorial and can be associated with stream marshes, brackish waters, epiphytes, humid soils and leaf litter (Riley, Rosa & Silveira, 2021). Adults are frequently found in trees, shrubs, and herbs (Lloyd, 2002). However, some firefly species have also been recorded opportunistically occupying urban areas, such as green spaces, backyards, urban parks, and vacant lots (Picchi et al., 2013; Lewis et al., 2020; Fallon et al., 2021). To date, most studies on fireflies have been done in natural and well-preserved habitats, while urban areas remain poorly studied. Therefore, more studies are needed on the diversity of these bioluminescent beetles in cities and peri-urban areas, as well as the threats and challenges posed to them in these modified landscapes.

In general, fireflies can be considered good bioindicators because they are highly diverse, have a relatively stable taxonomy, are associated to specific habitats and elevational ranges, and their populations sizes respond to environmental changes and different disturbance factors (e.g., temperature and artificial light at night; Hagen et al., 2015; Colares et al., 2021; Fallon et al., 2021). They also have an important ecosystem role as predators of soft bodied invertebrates, and as prey of birds, frogs, and bats (Lloyd, 1973; Fallon et al., 2021; Riley, Rosa & Silveira, 2021).

A total of 167 and more than 1,200 species have been documented from the Nearctic and Neotropical biogeographical regions respectively (Costa, 2000; Silveira et al., 2020; Fallon et al., 2021). Mexico, located in the transition of both regions, is a hotspot of fireflies, as it harbors more than 10% (284 species) of all described species of fireflies in the world (Pérez-Hernández, Zaragoza-Caballero & Romo-Galicia, 2022; Zaragoza-Caballero et al., 2023; Gutiérrez-Carranza, Domínguez-León & Rodríguez-Mirón, 2023; Gutiérrez-Carranza, Zaragoza-Caballero & Domínguez-León, 2023). Several studies on Mexican fireflies have been conducted in recent years, but none of them have considered urban assemblages. However, as occurs in other regions of the world, fireflies inhabiting urban or peri-urban Mexican localities are constantly faced with several threats associated with human disturbance and may even suffer frequent local extinctions in urban zones. Therefore, it is urgent to evaluate their conservation status and determine their current threats to generate conservation strategies.

Recently, based on anecdotal reports and records from citizen science platforms (Naturalista, 2022), we learned about the current presence of firefly populations within the urbanized area of Morelia, the capital city of Michoacán state, in central-western Mexico. We also compiled stories about the existence of vast firefly populations in the city from the 1950’s through the 1980s, which gradually decreased over the decades until they disappeared. However, to date no scientific reports have documented this phenomenon, and there are no specific conservation strategies or environmental protection for fireflies or their habitats. In addition, there are no studies on fireflies in Morelia, and only two species have been reported in the city before this work: Photuris versicolor (Fabricius, 1798) ((under Photuris trilineata (Say, 1835)) and Pyropyga minuta Gorham, 1881 (Zaragoza-Caballero, 1993). Note that Zaragoza-Caballero et al. (2023) also mentioned finding Photinus apahtzii Zaragoza-Caballero and López-Pérez, 2023 in the city, but the authors have since confirmed that the record should be treated as an error (S Zaragoza-Caballero and S López-Pérez, 2023, personal communication).

Morelia has shown a remarkable increase in size and human population since the 1950s, and its growth has become faster and less controlled in recent years, leading to rapid peri-urbanization that threatens the biodiversity that survives in the area (López et al., 2001; MacGregor-Fors, 2010; Ortiz & Vieyra, 2018). For instance, of the area currently covered by the city, in 1960, 93.1% was covered by different types of vegetated spaces and crops; by 1990 that percentage diminished to 28.1% and was less than 8% by 2020 (López et al., 2001; Bollo, Martín & Martínez, 2022). Despite that rapid transformation, there are few studies on the effects of urbanization on insect biodiversity in the region. These few examples have generally found dramatic effects. For instance, there was a drastic loss of lepidopteran diversity in highly urbanized sites from Morelia (Quiróz Pérez, 2008), while high lepidopteran diversity in peri-urban areas was associated with the presence of young trees and the coexistence of both native and introduced plants in those areas (López-López, 2011). As lepidopterans are bioindicators of insect diversity and environmental change (Cabrero-Sañudo et al., 2022), we expect other groups of insects to show similar patterns.

In the present work, we used recent records from citizen science platforms, entomological sampling, and entomological collections to determine the main stressors for urban firefly populations in Morelia. We determined the current diversity of fireflies from Morelia and characterized their habitats using different variables, such as terrain slope, elevation, landscape class, habitat size, type and proportion of vegetated spaces, and presence and size of watercourses. We also evaluated disturbance factors associated with those habitats, such as the degree of urbanization (urban, peri-urban, and extra-urban), light pollution levels, and type and proportion of urban surfaces. We also characterized sites where fireflies have been reported as extinct to compare them with sites where fireflies still survive.

Materials and Methods

Study area

Morelia municipality is in the central-western region of Mexico in the state of Michoacán (between 19°27′06″ and 19°50′12″N, −101°01′43″ and −101°30′32″W) (Fig. 1), at an average elevation of 1,920 m asl. The city currently covers an area of 157.6 km2 (IMPLAN, 2022). It has a temperate sub-humid climate type Cb(wo)(w)(i.)g with summer rains (June–September; Cigna & Tapete, 2022). The annual mean minimum temperature in the city is 10.5 °C, the annual mean maximum temperature is 26.9 °C, and the mean annual precipitation is 770.5 mm (National Weather Service, 2022). In the past decade, the maximum and average temperatures have increased by 0.02 °C and 0.07 °C per year, respectively, and the total annual precipitation has increased by about 7 mm per year (Cigna & Tapete, 2022).

Figure 1 Growth of the urban area and human population of Morelia city (Michoacán, Mexico) since 1960.

Based on López et al. (2001); López Núñez & Pedraza Marrón (2012); IMPLAN (2022). This OSM Standard map is made available under the Open Database License.

Land cover in Morelia includes temperate mixed forest, Eucalyptus plantings, wetlands, shrubs, and crops; land uses include residential, industrial, and commercial use, in addition to conservation and green areas (e.g., well-preserved woods, parks, open grasses, public gardens, vacant lots, and corridors mainly located in the Southern part of the city; López et al., 2001; IMPLAN, 2022). A number of seasonal and year-round creeks, rivers, and artificial channels cross or influence the urban area of Morelia; these include the Río Grande (previously known as Río Guayangareo), Río Chiquito, La Mintzita, and Presa Cointzio which are part of the Río Grande, Río Chiquito, and Los Pirules microwatersheds (López Núñez & Pedraza Marrón, 2012; IMPLAN, 2022). Morelia is situated in alluvial plains that contain very gentle slopes (27.5% of the total area), gentle slope–intermediate piedmont (32.5%), and steep slopes (16%) (López et al., 2001).

Morelia is in the Cuitzeo hydrological basin, between the Cuitzeo lagoon and Lake Pátzcuaro. According to López Núñez & Pedraza Marrón (2012), during the pre-colonization era, the area surrounding the city harbored numerous swamps which increased in size during the rainy season, and human populations settled in the peripheral areas, in the mountains. Later, during Spanish colonization from the XVI–XVIII centuries, swamps were replaced by agricultural and livestock zones, and homes. During the XIX century, the city had slow growth and a relatively stable population size. However, Morelia began to expand rapidly in the mid-twentieth century. By 1960, the population of Morelia reached 106,000 (double its size in the XIX century), and by 2020, it had reached nearly 850,000, mainly due to migration from rural areas and other urban areas (e.g., Mexico City) (Fig. 1; López et al., 2001; López Núñez & Pedraza Marrón, 2012; INEGI, 2022). As a result, the urbanized area of the city has mostly followed an “edge-expansion” growth model and increased in size by 506% from 1960 to 1990 and nearly doubled in area from 2000 to 2010 (López et al., 2001; López Núñez & Pedraza Marrón, 2012; Cigna & Tapete, 2022; INEGI, 2022). Thus, in 2020 Morelia City was approximately 15 times its size in 1960, with an average rate of urban expansion of 1.8 km2 or 1.6% per year (Fig. 1; López et al., 2001; López Núñez & Pedraza Marrón, 2012; Cigna & Tapete, 2022; INEGI, 2022; IMPLAN, 2022).

Distributional and taxonomic data on fireflies

Distributional data of the firefly populations in Morelia were obtained from: (i) anecdotal reports of recent and past sightings obtained from surveys and interviews; (ii) public databases and citizen science projects, such as Naturalista (2022) and GBIF (2021); (iii) direct observations and entomological sampling performed from 2020 to 2022 (Fig. 2, Table S1); and (iv) few specimens from the entomological collection of the Laboratorio de Entomología Biol. Sócrates Cisneros Paz, Universidad Michoacana de San Nicolás de Hidalgo (UMSNH). Most of the data were obtained between 2016 and 2022 and were used to determine the current habitat of fireflies in the city. Systematic entomological samplings were performed by using aerial nets for 4 h each night in specific localities, during the previous and posterior days of New Moon from May to September 2022 (Table S1); occasional, nonsystematic diurnal and nocturnal samplings were also done by using light traps, Malaise traps and direct collection on different sites throughout 2020–2022. We also used the date of the records from Naturalista to determine the entomological samplings of the less common species and consulted participants of Naturalista to corroborate and correct coordinates of some records (e.g., downtown). Sites where firefly populations were reported to be present in the past but where they have not been seen during the last decade, were defined as extinction sites. We characterized sites of extinct populations by comparing their characteristics with those of places where fireflies are still present.

Figure 2 Localities of extant (green squares, yellow dots and gray triangles) and extinct orange circles firefly populations were found in the urban area of Morelia city.

Based on records from 2016 to 2021 obtained in Naturalista (2022) and GBIF (2021), and entomological samplings made during 2022. Image ©2023 CNES/AIRBUS, Landsat/Copernicus, Image ©2023 Maxar Technologies. Map data ©2023 Google, INEGI.

Based on the specimens available from collections and samplings, and photographs from Naturalista (2022), we identified firefly genera, species, and morphospecies mainly using the works by Gorham (1885), Green (1961), Cicero (1988), Zaragoza-Caballero (1995a, 1995b, 1996, 2002, 2012), Lloyd (2002), and Zaragoza-Caballero et al. (2020, 2023). We also compared firefly specimens with type material deposited at the National Insect Collection at the Universidad Nacional Autónoma de México. The taxonomic identity of fireflies from Naturalista platform was determined to the highest level of taxonomic resolution (species) in cases of very distinctive fireflies from the assemblage; photographs of visually similar species (e.g., Photinus ater, P. parvusater, and other) or with low-resolution were treated at family or genus level and only temporal and spatial data of the presence of fireflies was considered. The Secretary of Environment and Natural Resources of the Government of Mexico granted the collection permit for the project “Surviving urbanization: Evaluation of the main anthropic pressure factors on firefly populations in the urbanized area of Morelia, Michoacán” (SGPA/DGVS/O5322/22). All specimens collected during the study were deposited at the Gene Library and Entomological Collection of the Behavioral Ecology Laboratory, at the Faculty of Biology, UMSNH (GCEUMSNH) and will be part of a reference collection of the Lampyridae of Michoacán.

Based on the preliminary data compiled, we added information on different habitat variables for each site to characterize fireflies’ habitats. These were: (i) geographic location (decimal coordinates, elevation), (ii) habitat characteristics (e.g., habitat size, landscape class, type and percentage of vegetated space, watercourses presence and features, terrain slope), and (iii) disturbance factors associated with urbanization (i.e., degree of urbanization, type of urban surfaces or built cover, percentage of urban surfaces, radiance value) (Table S2). All of these variables have been previously associated with ecological requirements and the main threats to different firefly life stages (Lloyd, 2002; Kazama et al., 2007; Lewis et al., 2020; Fallon et al., 2021).

Characterization of urban firefly habitat

The landscape in Morelia was primarily classified according to MacGregor-Fors (2011) into urban land use categories of “commercial”, “conservation”, “green space”, “residential” or “industrial.” Vegetated spaces were also categorized into forests or woods (possessing a tree layer), shrubs (bush layer), meadows (open grass), crop land, urban green spaces (e.g., backyards or domestic gardens, parks, open grass, sport areas, public gardens), vacant lots, corridors (e.g., median strips, green areas along rivers or between residential areas), hills, or conservation areas. Firefly habitat size (the area of the specific site where fireflies were recorded) was measured using GIS analysis and satellite imagery from Google Earth (2021); for open spaces (such as those of extra-urban zones), we limited the habitat size to the area within a 500-m radius circular buffer (0.79 km2) around the exact geographical point where fireflies were reported, in order to make all sites comparable. We also estimated the suitable area for fireflies in the zone as the percentage of vegetated area within a 500-m radius circular buffer zone around the center of each site of firefly observation. Then, we established three categories of the proportion of suitable habitat within the 500-m buffer: (1) highly vegetated (67–100% vegetated area), (2) moderately vegetated (34–66% vegetated area), and (3) sparsely vegetated (0–33% vegetated area).

Adult fireflies are terrestrial insects commonly associated with watercourses and wet habitats, and the larvae of some species are truly aquatic or facultatively aquatic (Lloyd, 2002; Jäch & Balke, 2008; Riley, Rosa & Silveira, 2021). To evaluate whether urban fireflies in Morelia are occupying habitats with those characteristics, we recorded the presence and type of watercourses (e.g., lakes, creeks, rivers, ponds, channels, and wetlands) in the vicinity of the firefly habitats, and whether those watercourses were seasonal or permanent. We also calculated the approximate watercourse width and the distance between the site of the firefly sighting to the nearest watercourse using GIS analysis and satellite imagery from Google Earth (2021).

Slope has been found to be related with the geological conditions that fireflies prefer; very gentle and gentle slopes have been previously reported to be more suitable for fireflies’ habitat because they generally have damp conditions with slow stream velocity and a high density of suspended sediments, which are beneficial for larval development (Kazama et al., 2007). The slope angle of each site was obtained from a Digital Elevation Model of Morelia (CONABIO National Commission for the Knowledge and Use of Biodiversity, 2023) using GIS analysis (QGIS 3.16.14-Hannover) and was characterized in four categories: flat plains to very gentle slopes (0−4% slope angle), gentle slopes (>4−8%), moderate slopes (>8−16%), and steep to extremely steep slopes (>16%).

Disturbance factors

Based on the urban gradient categories suggested by MacGregor-Fors (2010), firefly sites were sorted into (i) urban core, (ii) peri-urban zone, or (iii) extra-urban zone depending on the geographic location in the city. We also included an (iv) extinction zone, as a fourth category to compare with the sites of extant populations habitats. This categorization allowed us to estimate the degree of influence of urban development on the city’s ecosystems.

Urban surfaces in each locality were sorted into residential areas, streets and/or avenues, commercial buildings, parking lots, or industrial areas. This information was utilized as a surrogate of the intensity of urbanization (based on MacGregor-Fors, 2011). We also estimated the percentage of impervious urban surfaces within a 500-m radius circular buffer zone around the center of each site of firefly observation to analyze its relationship with the presence/absence of firefly populations. Following MacGregor-Fors (2011), the percentage of urban surfaces was sorted into three categories: (1) sparsely developed (0–33% built cover), (2) moderately developed (34–66% built cover), and (3) highly developed (67–100% built cover). All estimations and changes were calculated using GIS data analysis and satellite imagery available from Google Earth (2021).

Fireflies depend on their bioluminescence for feeding, defense against predators, and reproduction, making them highly vulnerable to light pollution (Stanger-Hall, Lloyd & Hillis, 2011; Owens & Lewis, 2021; Owens et al., 2022). To estimate the degree of light pollution to which fireflies in Morelia are exposed, we compiled data from the nighttime light maps produced by the Earth Observation Group (2021). Based on Vaz et al. (2021), we used the radiance values (nW/cm2/sr) data set provided by the Visible and Infrared Imaging Suite (VIIRS) and Day Night Band (DNB) satellites as an indirect measure of light pollution and as a proxy for urbanization for each site where fireflies were observed in Morelia in 2021. Each pixel in the light pollution map shows radiance values ranging from 0 to 63, where zero represents total darkness (often associated with rural zones), and 63 represents very bright sites associated with urban zones.

Statistical analysis

We tested for differences among firefly population sites in their (i) habitat size, (ii) proportion of urban surfaces, (iii) slope, and (iv) radiance values, as a function of their degree of urbanization (urban core, peri-urban zone, extra-urban zone, extinction zone) or land use category (residential, conservation, or green areas). Because the data showed a normal distribution and we have evidence of homoscedasticity, we performed these analyses using Analysis of Variance (ANOVA) followed by multiple comparisons adjusted with the Bonferroni correction, which is useful for sets of univariate non-independent analyses. All statistical analyses were performed with R 4.1.2 (R Core Team, 2021) using the package “vegan” (Oksanen et al., 2015).

Results

We documented firefly populations at 37 localities associated with the urban area of Morelia city and surrounding areas based on the different sources consulted (Fig. 2, Table 1, Table S2). A total of 86.5% (32 sites) of the records represented extant or possibly extant firefly populations, and the rest (13.5%, 5 sites) were reported as extinct populations (all documented through interviews and surveys); at least 40.5% (15) of the sites were exclusively recovered from citizen science platforms, 40.5% (15) were sampled, 5.4% (2) were recovered from GBIF, and the rest (5 sites of firefly extinctions) were exclusively recovered from anecdotes. Most firefly records were from within the past decade (from 2016 to 2022) through sampling collections performed for this work (56.7%) and citizen science platforms (21.6%); only a few records were obtained from anecdotal (13.5%) and public databases reports or entomological collections (8.2%).

Table 1 Firefly species associated with the urban area of Morelia city and surrounding areas, and characteristics of their habitats.

Species	Urban zone	Behavior and morphology	Habitat in Morelia	Elevation (m asl)	Source	
Lampyrinae						
Cratomorphini						
Aspisomoides bilineatum (Gorham, 1880)*	E, P	Nocturnal, fully winged females	Wetlands and open grasses, near creeks and rivers	1,999–2,153	ES	
Cratomorphus halffteri Zaragoza-Caballero, 2012**	E	Crepuscular (?); unknown females	Well preserved woods	2,200	ES, NT	
Pyractomena striatella Gorham, 1880**	CA, P	Nocturnal, fully winged females	Wetlands near channels and rivers	1,891–1,930	ES, NT	
Photinini						
Photinus acutiformis Zaragoza-Caballero and Cifuentes-Ruíz, 2023**	CA	Nocturnal, fully winged females (new record)	Wetlands near channels	1,898	ES	
Photinus anisodrilus Zaragoza-Caballero, 2007**	U, CA, P	Nocturnal, fully winged females	Open grasses, shrubs and wetlands near channels and lake	1,898–1,983	ES	
Photinus ater *	E, P	Diurnal	Shrubs, near rivers	2,362	ES	
Photinus barrerae Zaragoza-Caballero and Rodríguez-Mirón, 2023*	E	Crepuscular; unknown females	Well-preserved woods, often near creeks and rivers	2,153–2,200	ES, NT	
Photinus chipirietetsi Zaragoza-Caballero and Vega-Badillo, 2023**	P	Nocturnal; unknown females	Shrubs and open grasses near creeks	2,045	ES	
Photinus extensus Gorham, 1881**	E	Nocturnal; females with reduced wings (brachypterous)	Well-preserved woods	2,200	ES	
Photinus guillermodeltoroi Zaragoza-Caballero and Rodríguez-Mirón, 2023**	CA, P, E	Nocturnal; fully winged females (new record)	Well-preserved woods and open grasses near creeks and lakes	1,983–2,200	ES, NT	
Photinus leobonillai Zaragoza-Caballero and Domínguez-León 2023**	E	Nocturnal; unknown females	Well-preserved woods	2,200	EC (1991), NT	
Photinus parvusater Zaragoza-Caballero, 1995*	P, E	Diurnal, fully winged females	Well-preserved woods and shrubs often near rivers and ponds	1,983–2,200	EC (1994), ES, NT	
Photinus vegai Zaragoza-Caballero y Cifuentes-Ruiz, 2020**	E	Nocturnal; unknown females	Well-preserved woods	2,200	ES	
Photinus zuritai Zaragoza-Caballero and Cifuentes-Ruiz, 2023**	U, CA, P, E	Crepuscular, Nocturnal; fully winged females (new record)	Well-preserved woods, public gardens, and open grasses near different water bodies	1,904–2,200	ES	
Photinus sp. ca. brailovskyi Zaragoza-Caballero, 2017*	E	Diurnal; fully winged females	Well-preserved woods	2,200	ES	
Photinus sp. 1 (under P. pyralis in Zaragoza-Caballero et al., 2023)*	P	Nocturnal; females with reduced wings (brachypterous)	Shrubs and open grasses near creeks	1,999–2,045	ES	
Photinus sp. 2*	E	Diurnal	Well-preserved woods	2,010–2,200	NT	
Pyropyga alticola Green, 1961**	P, CA, U	Diurnal; fully winged females	Public gardens, open grasses, vacant lots	1,898–1,910	ES, NT	
Pyropyga minuta (LeConte, 1851)	U	Diurnal; fully winged females	Public gardens, open grasses	1,910	EC, ES, NT, ES	
Pyropyga nigricans (Say, 1823)*	P	Diurnal; fully winged females	Public gardens, open grasses	1,963	NT, ES	
Pleotomini						
Pleotomus emmiltos Zaragoza-Caballero, 2002**	E	Nocturnal; bioluminescent, flightless females (apterous; new record)	Well-preserved woods, open grasses	2,185–2,200	ES	
Pleotomus pallens Zaragoza-Caballero, 2002*	CA	Nocturnal; flightless females (apterous)	Well-preserved open grasses	1,983–1,999	ES	
Photurinae						
Photurini						
Photuris fulvipes (Blanchard, 1846)**	P, E	Crepuscular, nocturnal; fully winged predatory females	Well-preserved woods, and open grasses near channels and rivers	1,930–2,200	ES	
Photuris lugubris Gorham, 1881**	E	Nocturnal; fully winged predatory females	Well-preserved woods	2,200	ES	
Photuris group versicolor (Fabricius, 1798)*	U, CA, P	Nocturnal; fully winged, predatory females	Wetlands, open grasses, public gardens, near waterbodies	1,898–2,045	EC, ES	
Photuris sp.*	E, P, CA	Nocturnal; fully winged, predatory females	Well preserved woods, wetlands, and meadows near waterbodies	1,898–2,200	ES, NT	
Notes:

Urban zones: U, urban core; CA, conservation area within the urban core; P, peri-urban; E, extra-urban. Method of sampling: EC, entomological collection of the UMSNH (sampling year indicated in the table); ES, entomological sampling (2021, 2022); NT, Naturalista citizen science project (2016 to 2022).

* New records for Morelia.

** New records for Michoacán, mainly based on Zaragoza-Caballero et al. (2020, 2023) and Pérez-Hernández, Zaragoza-Caballero & Romo-Galicia (2022).

In total, we obtained 351 records and identified 7 genera and 26 species of fireflies in Morelia and the surrounding areas from 2016 to 2022 (Fig. 3, Table 1; also see the Naturalista project “Luciérnagas de Michoacán”, and the dataset Pérez-Hernández, Mendoza-Cuenca & Romo-Galicia (2023). Five species were recorded in the urban core, 11 in conservation areas within the urban core (such as the Universidad Latina de América, UNLA), 13 in the peri-urban area, and 16 in extra-urban zones. At least 12 species were collected in more than one zone of the city, another 10 species were exclusively found in one well-preserved extra-urban site (Tsíntani), three were exclusive to peri-urban areas, one to the urban core, and one to an urban conservation area (at the UNLA). The most species-rich genus was Photinus (14 species), followed by Photuris (4), Pyropyga (3), Pleotomus (2), Aspisomoides (1), Cratomorphus (1), and Pyractomena (1). A total of 12 firefly species (46.2%) were crepuscular or nocturnal, with adult females and males showing luminescent organs (Aspisomoides bilineatum, Pyractomena striatella, Photinus spp., Photuris spp.). Another five species have brachypterous bioluminescent females (Photinus extensus, Photinus sp. ca. pyralis) and bioluminescent males (Table 1). Only seven species (26.9%) are diurnal fireflies without luminescent organs in adults of either sex (all Pyropyga species, Photinus parvusater, Photinus sp. ca. brailovskyi, Photinus ater, Photinus sp. 2). Adult males of the two nocturnal Pleotomus (7.7%) are non-luminescent, while the females are luminescent larviform, apterous and flightless. The females of at least five (19.2%) nocturnal species are still unknown to date, and we reported here for the first time the females of Photinus acutiformis, P. guillermodeltoroi, P. zuritai, and Pleotomus emmiltos.

Figure 3 Urban fireflies of Morelia, Michoacán.

(A) Photinus anisodrilus; (B) P. zuritai; (C) P. parvusater female; (D) P. leobonillai; (E) P. guillermodeltoroi; (F) P. vegai; (G) P. barrerae; (H) P. acutiformis; (I) Pyractomena striatella; (J) Pleotomus emmiltos. Image credits: (A, B, C, G) Cisteil X. Pérez-Hernández; (D, E, F) Pablo Alarcón Chairés (D) https://www.naturalista.mx/photos/146942173, CC BY NC; (F) Pablo Alarcón Chairés, https://www.naturalista.mx/photos/90868076, CC BY NC; (J) Pablo Alarcón Chairés, https://www.naturalista.mx/photos/131149998, CC BY NC; (E) Ana M. Flores; (H, I) Danna Betsabe Rivera Ramírez.

Characterization of firefly habitats

Urban, peri-urban, and extra-urban sites with extant and possibly extant firefly populations were mainly distributed in the Central and Southern regions of Morelia, in a variety of vegetated spaces such as backyards, cultivated lands, hills, open grass, meadows, public gardens, residential areas, corridors, roads, and vacant lots distributed among the different urban zones (Fig. 4A; Table S2). The five sites where extinct populations were reported had public gardens (1 site) and green spaces within residential areas (4 sites). Firefly habitats were distributed in an elevational range of 1,886–2,384 m asl, with most sites found at the lower altitudes. There were significant differences in elevational range among the urban zones (F3,33 = 19.42, p < 0.001; Table S1, Fig. S1). Of the firefly habitat sites, 29.7% of sites were categorized as residential landscapes (including the five extinction sites), 43.3% as green spaces, and 27% as conservation zones; there were no reports of populations in industrial or commercial landscapes.

Figure 4 Variation in characteristics of firefly habitats among urban core, peri-urban, extra-urban, and firefly extinction zones of Morelia city, in 2021: (A) types of vegetated spaces; (B) habitat size; (C) proportion of suitable habitat; (D) types of watercourses near firefly habitats; (E) slope category. Also, (F) variation in firefly habitat size among three land use categories.

The size of the current firefly habitats (the total area of the specific site where fireflies have been observed) in the urban core and peri-urban zones ranged from 0.0005 km2 to 0.79 km2 (Fig. 4D), whereas in extra-urban zones, the size of firefly habitats varied from 0.46 km2 to 0.79 km2. There was a significant difference in habitat size between the extra-urban zones and the other two urban zone categories, as well as with the sites of extinct populations (F3,33 = 20.87, p < 0.001) (Fig. 4D). In addition, habitat size differed significantly among the different categories of land uses (F2,34 = 6.509, p < 0.001) (Fig. 4F). Based on the suitable habitat for fireflies (measured as the vegetated area in the vicinity of the firefly habitat), a total of 24% sites were sorted as highly vegetated, 28% as moderately vegetated, and 48% as sparsely vegetated, whereas the five sites with extinct populations showed moderate (two sites) or sparse proportions of vegetation (three sites) (Fig. 4B). All extra-urban sites were highly vegetated.

In total, 78.1% of the 32 sites with extant firefly populations were near watercourses, such as creeks, rivers, ponds, lakes, channels, wetlands, and wellsprings (Fig. 4C) with a width range of 1 m (creeks) to 500 m (lakes and wetlands), and a distance between 0 m and 350 m from the observation site to the nearest point of the watercourse. At least 59.4% were permanent watercourses, and 34.4% were longer than 500 m. For extinct populations, only one of the five sites was near a permanent river (longer than 500 m); the site was 242 m from the river. Most of the urban and peri-urban sites but only half of the extra-urban sites were near water bodies.

Also, 59.5% of the sites were on very gentle slopes (alluvial or flat plains), 21.6% on gentle slopes, 8.1% on moderate slopes, and the remaining 10.8% of the sites corresponded to steep slopes. All five extinct populations were distributed in flat plains. We found significant differences among urban zones (F3,33 = 9.786, p < 0.001) in their slope category, with extra-urban zones showing higher slopes than the other three zones (Fig. 4E); also, there were significant differences in slope among land-use categories (F2,34 = 4.125, p = 0.0249), with conservation sites showing significantly steeper slopes than green spaces and residential areas.

According to our observations, at least Aspisomoides bilineatum, Pyractomena striatella and Photinus acutiformis (Photinini) seem to be exclusively associated with wetlands or flooding areas, and the immature stages of the former two species have semiaquatic behavior. All of the other species recorded from Morelia and surrounding areas seemed to be terrestrial during their adult stage, which were associated with meadows, open grass, vacant lots, public gardens, roads and/or woods (Table 1).

Disturbance factors for urban fireflies

The 32 firefly assemblages were more frequently located in the peri-urban zone (48.6%) than in the urban core (32.4%, including two conservations sites) or the extra-urban zones (19%); in contrast, four out of five extinct assemblages were in the urban core. The percentage of impervious urban surfaces within the 500-m radius circular buffer ranged from 0 to 95.4%; a total of 53.1% of the sites were sparsely developed, 31.3% were moderately developed, and 15.6% were highly developed. Four of the five sites of extinct assemblages were highly developed. The proportion of urban surfaces was similar between urban core and extinction zones, which had higher values than in the peri- and extra-urban zones (F3,33 = 43.47, p < 0.001; Fig. 5A).

Figure 5 Disturbance factors influencing the firefly populations in Morelia city.

Variation in the (A) proportion of impervious urban surfaces within a 500-m buffer; and (B) light pollution levels (radiance) among urban core, peri-urban, extra-urban, and firefly extinction zones in 2021.

Radiance values in firefly habitats ranged from 0 to 41.5 nW/cm2/sr; in sites of firefly extinction, radiance varied from 22.3 to 34.4 nW/cm2/sr and showed a similar pattern to urban surface percentage. In peri-urban sites radiance was lower than 22.2 nW/cm2/sr and lower than 4.78 nW/cm2/sr in extra-urban sites. We found significant differences among urban zones (F3,33 = 43.65, p < 0.001) with the urban core and extinction zones showing similar values of light pollution (Fig. 5B). We did not find significant differences in light pollution levels among land use categories (F2,34 = 2.819, p = 0.073).

Discussion

According to anecdotal evidence, some decades ago there were vast populations of nocturnal fireflies in sites where the urban zone of Morelia city is now located. However, fireflies in the region have been poorly studied. Before this work, no systematic studies of fireflies had been carried out in the city, its surrounding areas, or even in the entire state of Michoacán. Our work represents the first effort to document the fireflies found in Morelia, in addition to characterizing their current habitats and identifying threats to their persistence in urban zones. We increased the firefly species list from only two previously documented species to 26 (a 92.3% increase) and found females of some species for the first time (which will be described in posterior taxonomical works). We also detected sites of local extinction, which differed from sites with current firefly populations mainly in that they both lack vegetated spaces, have high levels of urban development (i.e., low proportion of impervious surfaces) and high levels of light pollution; thus, it seems likely that these factors are strongly related to the local extinction of fireflies in sites where they had been present only a couple of decades ago.

Based on previous studies, a total of six genera and 16 firefly species had been reported in Michoacán state (Pérez-Hernández, Zaragoza-Caballero & Romo-Galicia, 2022); later six new Photinus species were described by Zaragoza-Caballero et al. (2023). According to our findings, the species list for Michoacán now contains at least nine genera (Cratomorphus and Pyractomena as new records) and 41 species (at least 14 new records for the state; Table S3). This nearly doubles the previous list for Michoacán, which now contains 14.6% of the total Mexican lampyrid fauna (284 species). Here, we highlight the relevance of citizen science platforms such as Naturalista and targeted entomological sampling as highly important data sources for inventorying current biodiversity. Without those sources, the current species list of fireflies from Morelia and Michoacán would be less complete. Citizen participation through interviews was not only relevant to increase the temporal scale of presence-absence data on fireflies but also to inform people on the conservation of insects and their habitats. However, our interviews implied a bias towards adult bioluminescent species, therefore future studies on urban fireflies based on citizen science could involve trained volunteers to perform diurnal surveys.

Fireflies were mainly found in the Central and Southern regions of Morelia, perhaps because those areas have been the most recently developed and still show moderate to high levels of vegetation as well as high levels of humidity throughout the year (IMPLAN, 2022). These areas were previously crop fields or conservation areas and were only recently changed to residential use. As we expected, fireflies in Morelia are associated with a variety of vegetated spaces in three different land covers and are more frequent in extra-urban and peri-urban green spaces and conservation zones where the environmental conditions seem to be better for them. These favorable habitat conditions can be summarized as larger, moderately to highly vegetated habitats near water bodies with low levels of urban development and low light pollution. In contrast, in Morelia fireflies are less frequent in residential areas and were never detected in industrial or commercial lands. Residential areas are usually sparsely vegetated, highly developed, and show high levels of light pollution, which are very similar to conditions in sites where fireflies are already extinct. Previous studies have also reported a negative impact of urbanization on lampyrids and other bioluminescent coleopterans of Elateroidea from Brazil and suggested that their disappearance or decrease in urbanized sites could be associated with a reduction of green spaces and increase in artificial illumination (Viviani, 2001; Viviani, Rocha & Hagen, 2010). Meanwhile, Fallon et al. (2021) found that habitat loss, light pollution and climate change are the main threats for at least 66.6% of North American firefly species, and Picchi et al. (2013) also found a negative relationship between the abundance of two firefly species and the level of urbanization of their habitats in Turin, Italy.

Although we found firefly habitats in Morelia that could still be large enough to harbor healthy populations of a variety of species, most are very small in size. For instance, the smallest habitat size for extant firefly populations in the city was about 0.0005 km2 (an urban garden near a creek in the urban core) where very small populations of two species have been recorded (Photinus guillermodeltoroi and Photuris versicolor). In particular, P. guillermodeltoroi was almost exclusively found in peri-urban sites and therefore we suggest that that specific population from the urban core could be at high risk in such a small habitat. Lee et al. (2019) reported a minimum size of suitable habitat of 0.0039 km2 for terrestrial fireflies from Daegu and Gyeongbuk provinces in South Korea. In contrast, Picchi et al. (2013) reported green areas greater than 0.023 km2 for two species of fireflies from the city of Turin, Italy. Further efforts should be made to determine the minimum size habitat for each of the 26 species reported in this work and analyze the assemblage composition to better understand both their ecological requirements and threats, especially because only 8% of the current area of the city is still covered by green areas and crops where fireflies and other insects could survive in the long term (Bollo, Martín & Martínez, 2022).

Sites from the urban core of Morelia with extant firefly populations share several characteristics with sites where fireflies are already extinct. Hence, we could expect those firefly populations to decline if the environmental health in those places does not improve. In addition, peri-urban areas of the city are rapidly changing due to human activities and could soon be transformed in ways that make them more similar to the urban core if no sustainable planning is applied in the short term. For instance, the size of firefly habitats in peri-urban areas are not significantly different from sites in the urban core or extinction zones (Fig. 4B) and already show moderate levels of development and light pollution (Figs. 5A, 5B). This finding suggests that the current development of peri-urban sites could follow a similar model to the urbanization that created the current urban core. However, we are certain that many fireflies and other native insect populations that still survive within the urban core could recover their populations if the government and the public increase their efforts to diminish light pollution and increase the size of green areas.

As we expected, most of the firefly populations in Morelia are associated with a diversity of water bodies. The presence, seasonality, and width of watercourses varied widely among urban zones, and we did not detect any pattern among them other than most being permanent. However, most extinction sites do not have any waterbodies, which could also be a relevant factor explaining the absence of fireflies in those sites. In addition, we did not explore other factors associated with waterbodies that could be relevant for firefly survival, such as pollutants, high proportions of aquatic vegetation, and draining or channelizing waterbodies, which could directly cause firefly larvae mortality due to poisoning or habitat loss, or indirectly, through the loss of their prey (Takeda et al., 2006; Fauzdi et al., 2021). Future studies could evaluate the effect of pollutants and the presence of riparian vegetation on the population dynamics of fireflies.

Slope has been found to be a relevant variable of fireflies’ habitats because larvae are frequent on soggy ground and around ponds, marshes and streams which are usually found in flat plains or very gentle slopes (Lloyd, 2002; Kazama et al., 2007; Lee et al., 2019). Therefore, we expected that fireflies from Morelia would also be more frequent in flat plains, which cover about half of the city and used to be rice paddy fields (López et al., 2001; López Núñez & Pedraza Marrón, 2012). However, flat plains have also been attractive for crops, Eucalyptus plantations, and urban settlement beginning in the 1970’s, and currently the city continues to increase in areas with steeper slopes because of population growth (López et al., 2001; Bollo, Martín & Martínez, 2022). Current populations of fireflies and other fauna are likely more frequent in peri-urban and extra-urban areas because their slope degree is higher and are of less interest for human settlements. However, since human population rapidly grows and available spaces diminish at the same rate, we could expect that those steep zones of the city would also be urbanized, with negative impacts on ecosystem health and biodiversity if green spaces and conservation zones are not explicitly included in urban planning and development. Additional variables associated with the degree of inclination are its geographic orientation (i.e., windward or leeward), wind direction, and speed, which influence temperature and humidity levels and, therefore, fireflies’ horizontal distribution and reproductive behavior (Jaikla et al., 2020).

Habitat loss due to urbanization is one of the main threats to fireflies on a global and regional scale and has especially strong negative impacts on habitat specialist fireflies (Fallon et al., 2021). Based on our results, the proportion of artificial surfaces and light pollution (measured here as radiance) are the main stressors for fireflies from Morelia, similar to other studies (e.g., Picchi et al., 2013; Fallon et al., 2021; Vaz et al., 2021). Artificial surfaces are replacing vegetated areas at a high rate in the city, and their proportions are so high that only 1% of the total urban area is currently covered by green spaces. This means that only 1.6 m2 of green space is available per inhabitant in Morelia, despite the World Organization of Health recommendation of 9 m2 per inhabitant (Bollo, Martín & Martínez, 2022; IMPLAN, 2022). Thus, there are not enough vegetated spaces for fireflies and other insects to survive. On the other hand, light pollution is strongly associated with urbanization and is one of the main threats to nocturnal, crepuscular, and diurnal fireflies (e.g., by affecting sexual communication in bioluminescent adults, and prolonging daily and seasonal activity, and delaying dormancy in diurnal species; Viviani, Rocha & Hagen, 2010; Owens et al., 2020, 2022). Excess artificial light influences life story traits of fireflies, their assemblage structures, and the population dynamics and dispersal of other insect species (Khattar et al., 2022; Owens et al., 2022). In Morelia, we found high levels of light pollution derived from the increase in artificial light at night (ALAN), which likely has similar impacts on local firefly populations.

Land cover has been found to have a strong influence on insect species richness, abundance, and community structure, leading to poorer diversity in highly urbanized areas compared to peri-urban and extra-urban zones (Tzortzakaki et al., 2019). Urbanization not only provokes habitat loss but likely influences other variables associated with habitat suitability for fireflies and other insects, such as temperature and humidity levels, and plant composition (Adams et al., 2020). For instance, the meteorological station from downtown Morelia reports the lowest levels of precipitation and the highest monthly temperatures in the city, which has been directly associated with the accelerated growth of Morelia (IMPLAN, 2022). Picchi et al. (2013) also mentioned isolation and fragmentation of green spaces due to urbanization as relevant factors influencing colonization by viable populations of fireflies. Further efforts should be made to analyze the relationships between diversity and community structure of fireflies from Morelia with land cover and development levels. Other factors that could be explored in future studies are temperature, humidity, vegetation type, dew point, and heat index which have been associated with firefly abundances and sex proportion (Ramírez-Manzano et al., 2023).

Based on our findings, we suggest that the number of firefly species and their abundance within the city could rapidly decrease in the coming years due to the increasing effect of their stressor factors (except, perhaps, within conservation areas). Some inhabitants of Morelia have already experienced the loss of fireflies and a decrease in their abundance in recent years, and they even notice the potential species substitution after land use changes. One anecdote supports this hypothesis: “fireflies used to shine in orange color, but they changed to green when the forest was cut, and meadows and buildings replaced it” (María Eloísa León García, 84 years old; 2022, personal communication). The species that was initially observed could be Photinus barrerae, which emits an amber colored bioluminescence and is only associated with well-preserved woods in extra-urban zones from Morelia; it may have been replaced by Photuris versicolor, which is a more resistant species that emits green light and still has high abundance in the same site, which is now a school campus with large areas of introduced grass.

As other studies have suggested, some species seem to be well adapted to urban areas and tolerate higher levels of light pollution (Viviani, 2001; Viviani, Rocha & Hagen, 2010). In Morelia, at least Photuris versicolor, Photinus zuritai, and all of the Pyropyga species found seem to be highly resistant and are able to thrive in urban gardens from highly developed zones; Photuris versicolor and Photinus zuritai were present at high abundances. In contrast, other species (likely habitat specialists) seem to barely survive in peri-urban zones and are constantly exposed to habitat loss and other stressors derived from human activities (e.g., Pyractomena striatella, P. chipirietetsi, P. guillermodeltoroi); whereas P. acutiformis was only found in well-preserved wetlands in a conservation area with intermediate levels of light pollution in the urban core. Furthermore, some species could already be excluded from their original habitats within Morelia; for instance, Photinus leobonillai was collected in the city in 1991, but we did not find any specimens in the urban core during our 2021–2022 samplings, although it is reported in areas surrounding Morelia (Naturalista, 2022). Also, P. barrerae seems to be currently restricted to well-preserved forests with low levels of light pollution, a type of habitat that used to be widely distributed around the current periphery of the city. It seems that the species that are the most strongly affected will be crepuscular or nocturnal species whose females have low dispersal capacity (i.e., brachypterous or apterous females). In addition, even the more resistant species in Morelia were not able to inhabit sites with radiance values higher than 36 nW/cm2/sr.

In contrast, sites with conservation and restoration programs have shown the persistence of firefly assemblages or even an increase in their richness and abundances in the last two decades. That is the case for the Universidad Latina de América, an ecological conservation area within the urban core, and for Tsíntani, an extra-urban site and Voluntary Conservation Area, that used to be covered with croplands 20 years ago. This highlights the need for more patches and corridors of green areas with native plants, non-polluted water bodies and low levels of light pollution to preserve nocturnal wildlife and their contributions for people. Monitoring well-preserved firefly populations as well as those threatened by urbanization will be essential for the understanding of firefly population dynamics in the short and long terms in the cities in development.

Conclusions

Due to their sensitivity to environmental degradation, fireflies can be used as bioindicators of ecosystems health (Picchi et al., 2013; Hagen et al., 2015). Based on our findings, sites from the urban core and extinction sites in Morelia likely have the highest levels of environmental degradation, while peri-urban sites could face the same scenarios in the near future if environmental planning and conservation actions are not taken soon. On other hand, firefly fauna from Morelia is still quite rich in species, representing an important proportion of Mexican fireflies (9.28%), and fireflies are highly appreciated by some people in the city. Sharing their experiences with fireflies and our academic findings through science communication strategies would support potential conservation programs and restoration planning at sites where fireflies are now living on the edge of extinction. Cities are rapidly expanding, and peri-urbanization is becoming more frequent; in those scenarios, fireflies could be used as good bioindicators of environmental health that could help to detect and monitor zones in need of protection for nocturnal biodiversity and human health.

Supplemental Information

Supplemental Information 1 Sites from Morelia city and surrounding areas where firefly populations have been reported in the last decades.

Click here for additional data file.

CXPH thanks to David Venegas Suárez Peredo, Danna Betsabé Rivera Ramírez, Olivia Huerta Luna, Yuritzi Román Garibay, and Rodolfo Gutiérrez Bolaños for their great assistance in the entomological samplings in 2022. Special thanks to Pablo Alarcón, Adriana Acosta and members of “Tsíntani, Voluntary Conservation Area” for providing access to their site, entomological material, and photographs of fireflies. We are grateful to the Universidad Latina de América (UNLA), Instituto de Investigaciones sobre los Recursos Naturales (INIRENA), Botanical Garden of the UMSNH, Restaurante Los Laureles in La Mintzita, Municipal Pantheon of Morelia, Tres Cascadas Ecotourism site, Rancho La Planta, and all the people who provided access to their green spaces for firefly sampling and those who shared anecdotes on their experiences with Morelian fireflies. We also thank the UMSNH Entomological Collection for providing entomological material through Javier Ponce Saavedra, and the National Insect Collection at the Universidad Nacional Autónoma de México for access to the firefly type material through Santiago Zaragoza Caballero. Our manuscript was proofread by professional English translator Lynna Kiere.

Additional Information and Declarations

Competing Interests

Author Contributions

Field Study Permissions

Data Availability

The authors declare that they have no competing interests.

Cisteil X. Pérez-Hernández conceived and designed the experiments, performed the experiments, analyzed the data, prepared figures and/or tables, authored or reviewed drafts of the article, and approved the final draft.

Ana María Gutiérrez Mancillas conceived and designed the experiments, performed the experiments, analyzed the data, prepared figures and/or tables, authored or reviewed drafts of the article, and approved the final draft.

Ek del-Val conceived and designed the experiments, performed the experiments, analyzed the data, prepared figures and/or tables, authored or reviewed drafts of the article, and approved the final draft.

Luis Mendoza-Cuenca conceived and designed the experiments, performed the experiments, analyzed the data, prepared figures and/or tables, authored or reviewed drafts of the article, and approved the final draft.

The following information was supplied relating to field study approvals (i.e., approving body and any reference numbers):

The Secretary of Environment and Natural Resources of the Government of Mexico granted the collection permit for the project “Surviving urbanization: Evaluation of the main anthropic pressure factors on firefly populations in the urbanized area of Morelia, Michoacán” (SGPA/DGVS/O5322/22).

The following information was supplied regarding data availability:

The data is available at OSF: Mendoza-Cuenca, Luis. 2023. “Urban Fireflies (Coleoptera, Lampyridae) in Morelia, Michoacán, Mexico.” OSF. November 18. osf.io/2nam6.

Pérez-Hernández CX, Mendoza-Cuenca LF, Romo-Galicia A. 2023. Dataset of the Lampyridae (Coleoptera from Morelia, Michoacán, México. Universidad Michoacana de San Nicolás de Hidalgo Facultad de Biología. Available at https://doi.org/10.15468/3asucg (accessed 26 september 2023).

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
