# Peer review of "Living on the edge: urban fireflies (Coleoptera, Lampyridae) in Morelia, Michoacán, Mexico"

_PeerJ, doi:10.7717/peerj.16622_

## Round 0.1 · original submission · Minor Revisions

Dear Dr. Mendoza-Cuenca,

After the independent evaluation of three independent reviewers, all of them believe your manuscript is worthy of publication in PeerJ after a minor review. Please consider all issues raised by the reviewers and proceed to improve your text. Please do not forget to prepare a rebuttal letter informing of all issues that were corrected and justifying the maintenance of those you disagree with the reviewers.

Sincerely,
Daniel Silva

·

Basic reporting

The language in which this manuscript is written is clear, professional, and unambiguous. It is good that your manuscript is carefully checked. The abstract is well-written, unambiguous, and concise. The introduction is well-written, presents clear and concrete information, is coherent, and has a very appropriate sequence of ideas. The references used are adequate and primarily up-to-date. The materials and methods section is well presented, but it could be improved in the aspects mentioned in the Additional Comments section. Results. This section presents a synthesis of a large amount of information, which is a very valuable work and for which the authors have taken the time to synthesize it clearly and orderly. The discussion section is good but needs to be more focused (too long). Since this manuscript emphasizes four main aspects: determine the current diversity of fireflies of Morelia, Michoacán, characterize their current habitats, determine the main stressors, and characterize possible extinction sites for urban firefly populations, it is expected to explore or discuss the direct evidence. The discussion is the section that must be attended to closely because many results need to be addressed in detail without losing the common thread. The general structure conforms to PeerJ standards. The figures are relevant, but some need to be improved (see comments in additional comments section).

Experimental design

This work constitutes novel research within the aims and scope of the journal. The research question was well-defined and identified a knowledge gap in studying fireflies in Mexico. This research complies with high technical and ethical standards. The methods are described with sufficient detail and information to replicate.

Validity of the findings

The results of this work follow a logical structure, the data have been adequately analyzed, and the information presented in an orderly manner. The data are statistically sound. This work provides important and novel information about the diversity and new records of the family Lampyridae in the state of Michoacán. It should be noted that citizen science information and selective entomological sampling are essential and valuable sources of data for this type of research. The conclusions are in line with the objectives of the work.

Additional comments

The peri-urbanization increases the ecotone or peri-urban zone. This phenomenon occurs in the periphery of the cities and implies land-cover change affecting the balance of natural ecosystems (changes the composition of the insect community). It has been reported that firefly populations are declining due to human activities (e.g., habitat loss, light pollution, and climate change). In the manuscript "Living on the Edge: urban fireflies (Coleoptera, Lampyridae) in Morelia, Michoacán, Mexico," authors have tried to determine the current diversity, characterize their current habitats, determine the main stressors, and characterize possible extinction sites for urban firefly populations. This work is noteworthy because authors use data from different sources, such as records from citizen science platforms (usually not considered), entomological sampling, and entomological collections. As written, the manuscript is acceptable for PeerJ publication. However, it could be improved on the following issues:

- Lines 33- 34: The authors mention that fireflies are considered bioindicators. In the introduction section, could you explain why fireflies are good bioindicators? What other factors make fireflies considered bioindicators (apart from sensitivity to environmental changes)?

- The ideas in lines 85-110 could be synthesized. The objectives are very well stated.

- Lines 142-158: Although they refer to the study site as having more background, how relevant is the information presented in these lines? It could be summarized.

- Fig. 1: The labels of the localities inside the map cannot be distinguished.

- Fig. 2: The shading color of the "urban core area 2021" could be changed (e.g., light grey) to be able to distinguish better.

- Lines 172-173: The taxonomic identity of the Naturalista reports (2022) was performed to what taxonomic level? Is it possible to get to the species level with photographs? Explain briefly how the protocol was made with the records from this source.

- Consider deleting information from lines 223-227

- Line 274: In total, how many firefly records were used in the development of this study?

- Line 267: In Table 1, check the style when citing the author of the species (some have parentheses, others do not). The author could go in another column.

- Lines 274-292: could be synthesized and unified in one paragraph.

- Lines 299: Edit Fig. 4 so that the legend does not overlap the graphic. Change the layout of plots a), c), and d) with a single legend for all three.

- Lines 294-337: These results could be written in a much more concise form; the values of the statistics could be presented in a table.

Line 302: <0.001 is sufficient (check other values presented). Use a maximum of 3 decimal places.

Line 406: "That" repeated.

Lines 434-437: Future studies could evaluate the effect of certain pollutants on the population dynamics of these insects, as well as variables such as the presence of riparian vegetation, which has been reported to be of utmost importance for organisms associated with water sources.

Reviewer 2 ·

Basic reporting

Clear, unambiguous and professional English language is used consistently. The writing is clear and explicit.
Overall, the literature is well referenced and relevant. It is suggested to incorporate some citations (included in the comments).
Article structure, figures, tables, etc. meet publication standards

Experimental design

The work was rigorously done, which can be replicated in any city, and gathers data for the first time on fireflies for the city of Morelia. With the data provided, proposals can be made to halt urban developments and/or contemplate the establishment of green areas.
Research question well defined, relevant & meaningful / YES
Rigorous investigation performed to a high technical & ethical standard / YES
Methods described with sufficient detail & information to replicate / YES

Validity of the findings

Please highlight the relevance of the use of citizen science data (Naturalista) and consider for introduction and discussion.
Data robust, statistically sound, & controlled? / YES
Conclusions are well stated, linked to original research question & limited to supporting results / Overall YES

Annotated reviews are not available for download in order to protect the identity of reviewers who chose to remain anonymous.

·

Basic reporting

This study is a solid and necessary addition to the literature on firefly ecology and distributions, with important bearings on their conservation. The authors used excellent professional English throughout their text, and handsomely subsidized their writing with updated references (unless otherwise noted in my PDF). They developed a very clever approach (oriented by anecdotes) to clearly map and determine the correlates of firefly extinction at urban areas. This study also significantly improve access to knowledge on firefly distributions in the MIchoacán area helping to overcome important Wallacean shortfalls. Therefore, the article met PeerJ standards, and I strongly encourage the publication of this study after addressing some minor recommendations.

Experimental design

In my opinion, the article met PeerJ standards. The authors made use of solid and refined methods to map and determine the correlates of fireflies occurrences and extinctions. I have made minor recommendations in my PDF file, suggesting that authors better describe their survey methods, as well as how they addressed a potential a bias towards flashing/glowing species in people's anecdotes.

Validity of the findings

The findings in this study are solid and clear, and a necessary addition to the literature who will help inform baselines and standards for generations to come. The conclusions are well stated and based on the findings of this study.

---

## Round 0.2 · accepted · Accept

Dear Dr. Mendoza-Cuenca,

I am pleased to inform you that your manuscript has been formally accepted for publication in PeerJ. Congratulations on your hard work!

Best regards,
Daniel Silva

·

Basic reporting

The authors have demonstrated commendable diligence in comprehensively addressing my comments in this revised version. They have effectively acknowledged and incorporated suggestions pertinent to the paper's overarching theme. The Materials Methods and Results sections have been refined and presented. The figures have undergone substantial improvement. However, it is imperative to highlight that specific responses from the authors have led to inaccuracies in the text. For instance, in response to comment 8, the authors indicated changes in lines 331-333; however, the accurate correction should be applied to lines 305-307. Attention to such details is crucial in future revisions

The discussion section is thoroughly elaborated; nonetheless, I posit that its length, spanning 13 paragraphs, including the conclusions, could be succinctly summarized from my perspective.

Despite these concerns, I do not have reservations in endorsing the acceptance of this manuscript for publication in PeerJ.

Experimental design

No comment.

Validity of the findings

No comment.

Additional comments

No comment.